

# Microplastic contamination in canned fish sold in Türkiye

Sedat Gündoğdu[1] and Ali Riza Köşker[2]

[1] Faculty of Fisheries/Department of Basic Science, Cukurova University, Adana, Türkiye
[2] Faculty of Fisheries/Department of Fisheries and Seafood Processing Technology, Cukurova University, Adana, Türkiye

## ABSTRACT

The presence of microplastics (MPs) in processed seafood is a growing concern. In this study, 33 different canned fish brands belonging to seven producers were purchased from the Turkish market and investigated. MPs composition, possible sources, and potential intake were assessed. Light microscopy was used to quantify potential MPs, and micro-Raman microscopy was used to identify the polymer types. The results showed that all the samples had at least one MPs particle, and fragments were the most abundant (57.3%) shape of MPs. Polyolefin (21.88%) was the most common polymer type. The results showed that packaging and the production processes are the main possible sources of MPs. Human intake estimation risk is relatively lower since canned fish consumption is relatively low. The findings suggest that the risk related to MPs in canned fish should be considered one of the components of food safety management systems.

## INTRODUCTION

Worldwide plastic production has continuously increased and reached 367 million tons in 2020 (*Europe Plastics, 2021*). Massive production makes plastics a ubiquitous environmental pollutant (*Bergmann et al., 2022*). Plastic pollutants constitute an environmental threat due to their effects on various organisms (*Tekman et al., 2022*). The long persistence of plastic constitutes a major issue that enhances its negative environmental and economic consequences (*Mihai et al., 2022*).

Plastic pollutants can be classified according to their sizes as megaplastic (>100 mm), macroplastic (>20 mm), mesoplastic (20–5 mm), microplastic (5 mm–1 μm) and nanoplastics (<1 μm) (*GESAMP, 2019*). Microplastics (MPs) can be produced in certain sizes and forms (primary MP), which can also be formed by the physical, chemical, and biological degradation of larger plastics (secondary MP). Primary MPs are purposefully manufactured to produce (pre-production pellets) plastic items (*Gündoğdu et al., 2022b*) or applications such as personal care products (*Sun, Ren & Ni, 2020*). Secondary MPs are formed by breaking up large plastic items into small particles by various forces. MPs can be further classified based on their physical form as microfibers, microfilms, microparticles, and microbeads. MPs are a mixture of heterogeneous plastic particles with different shapes

Corresponding author
Sedat Gündoğdu,
sgundogdu@cu.edu.tr

(*Salerno et al., 2021*) and readily adsorb toxic pollutants (*Fu et al., 2021*). Moreover, MPs can remain in the environment for a long time (*De Falco et al., 2018*), and their persistent character attracted researchers to focus on the behavior of MPs globally (*Wang et al., 2021*).

MPs are ubiquitous in the environment, and they were found in the air (*Dris, Gasperi & Tassin, 2018*), drinking water (*De Frond et al., 2022*), soil and sediments (*Gündoğdu et al., 2022a*; *Yıldız et al., 2022*), freshwater and seawater (*Mihai et al., 2022*), and aquatic and terrestrial organisms (*Hale et al., 2020*). The ubiquity of MPs also diversifies their entry routes into the food chain (*Akoueson et al., 2020*). The main entry route of MPs into the food chain is mainly through aquatic food products. These food products range from table salt (*Gündoğdu, 2018*) to fish (*Blankson et al., 2022*), mussels (*Gedik, Eryaşar & Gözler, 2022*), and seaweeds (*Li et al., 2020*). However, MPs can enter the body of many organisms, especially humans, through direct inhalation (*Kashfi et al., 2022*), leakage from the package into food (*Sobhani et al., 2020*), and even though plastic cutting boards (*Habib et al., 2022*).

The wide usage of plastic as packaging also increases the risk of MP exposure by migrating MP from package to food. Moreover, processed packaged foods widely consumed during the COVID-19 period (*Oliveira et al., 2021*) increase the risk of MP exposure (*Deng et al., 2022*). Packing food with plastic or composite materials, including plastic, is a standard food preservation method. This method is also widely used for seafood. The most common packaging for seafood is canning. However, packages with different characteristics have been used in seafood and canned food packaging over time. It is worth noting that canning seafood, especially fish, including canning, has many advantages. Although the canning of fish may affect the quality of the final product, canned fish is still a good source of protein, vitamins, minerals, and fatty acids. Through canning, fish can be preserved for extended periods and transported to farther points. However, with this advantage, possible contamination in canned fish raises some concerns for public health. The most important of these concerns are bacterial load (*Hariri, Bouchriti & Bengueddour, 2018*), heavy metal pollution (*Mansouri et al., 2021*), and microplastic contamination (*Karami et al., 2018*; *Akhbarizadeh et al., 2020*; *Diaz-Basantes et al., 2022*). Despite these concerns, canned seafood is a popular food source worldwide because it is practical and inexpensive. The changing eating behavior and interest of consumers during the COVID-19 pandemic have also increased the preference for canned fish (*Nguyen Ngoc & Kriengsinyos, 2022*).

Canned seafood produced in Türkiye is exported to Europe and is also widely consumed by Turkish consumers. Although there is no reliable data source, approximately 33,000 tons of sea fish caught in 2019 were processed as canned. In addition, a significant amount of imported fish is consumed in the domestic market as canned. Some images of canned fish production in Türkiye, which are reflected in the media, raise concerns that food safety can be overlooked in the production processes, which may lead to contamination (*Haberturk, 2021*). Although different methods can be followed in various countries in the preparation of canned fish, in general, similar processing steps are generally applied for production. According to these process steps, making canned fish starts with the catch and is then transported to the processing facility, usually in the cold chain. Fish are thawed, gutted, and washed in transferred industrial plants to eliminate habitat contamination. These processes are followed by manual cleaning, where the skin, spines, and fat are removed.

Then, the washing process is carried out, and the next stage, the cooking process, is started. Then, various spices, salt, water, and oil are added according to the product type (*Karami et al., 2018*; *Akhbarizadeh et al., 2020*; *Diaz-Basantes et al., 2022*). Since no precautions are taken regarding MP pollution in these processes, contamination from the process is quite possible. The only concern related to fish canning is not rising due to product safety standards. In addition, marine pollution can also be transferred to the final consumer through fish. It seems unlikely that canned fish will transfer marine pollution to the end consumer since only the fillet of the fish is used. However, studies show that microplastics of certain sizes can also pass into muscle tissue. Therefore, marine plastic pollution can reach the end consumer in this way. This is the case both for microplastic pollutants and for other contaminants.

Although MPs in aquatic organisms have been investigated in recent studies (*Gündoğdu, Çevik & Ataş, 2020*; *Gedik & Eryaşar, 2020*), the occurrence of MPs in processed seafood products such as canned fish was rarely studied. To the best of our knowledge, only *Karami et al. (2018)*; *Akhbarizadeh et al. (2020)*; *Diaz-Basantes et al. (2022)* studied the presence of plastics in canned fish. *Karami et al. (2018)* considered the presence of meso- and microplastics in different brands of canned sardines and sprats purchased from other countries. *Akhbarizadeh et al. (2020)* investigated canned fish (tuna and mackerel) samples from the Iranian market, and *Diaz-Basantes et al. (2022)* investigated the most common canned fish brands sold in supermarkets in Ecuador. However, there is no study on MP particles in canned fish in the Turkish market. Hence, the aims of this study were: (i) to investigate the presence of MPs in almost all brands of canned fish available in the Turkish market and their characteristics such as color, size, type, and polymer composition, and (ii) to determine the possible correlations between the package type and MPs' content of canned fish, and (iii) assess the human intake of MPs through canned fish consumption.

## MATERIALS & METHODS

In total, 33 canned fish samples (with three replicates) belonging to 13 different brands were purchased from Turkish markets during the summer of 2021. The canned fish were produced by seven different companies for various brands. Fish species were Black Sea anchovy, Norwegian salmon, longtail tuna, yellowfin tuna, skipjack, and mackerel fish (Table 1). Table 1 represents all the characteristics (*i.e.*, the oil types, fish species, and salt concentration) of sampled canned fish. As shown in Table 1 that three different oil is used for additives (Canola oil, olive oil, and sunflower oil). Fish species in most canned fish were tuna fish ($n = 23$), and most were packaged in metal cans.

The MP extraction from canned fish was done according to the method applied by *Gündoğdu et al. (2021)*. Specifically, 50 g of homogenized sample from each can was weighted using a four-digit microbalance and then placed in a pre-cleaned beaker. The beakers were subsequently closed with aluminum foils in case of airborne contamination. For organic material digestion, a 30% of KOH: NaClO solution (a mixture of 700 mL of micro-filtrated water, 150 mL of saturated KOH solution (1,120 g/L), and 150 mL of NaClO with 14% active chlorine) was used (*Gündoğdu et al., 2021*). Then, 250 ml of the

**Table 1  Characteristics of sampled canned fish. In the code column, the first letters indicate the main production company; the second letters indicate brands, and the numbers indicate products.**

| Code | Party-no | Fish species | Additive oil | Other additives | Product weight | Package type |
|---|---|---|---|---|---|---|
| D-C-1 | 20848E | Tuna | Sunflower oil | Salt | 80 gr | Can |
| D-D-1 | 19/10/2020 | Yellowfin Tuna | Olive oil | Salt | 75 gr | Can (BPA free) |
| D-D-2 | 07-09-2020 | Skipjack | Sunflower oil, Canola oil | Salt | 80 gr | Can |
| D-D-3 | 05-10-2020 | Yellowfin Tuna | None | Water | 75 gr | Can (BPA free) |
| D-D-4 | 16.11.2024 | Norwegian Salmon | Olive oil | Salt | 100 gr | Aluminum |
| D-D-5 | 10-06-2023 | Skipjack | Olive oil | Salt | 125 gr | Aluminum |
| D-D-6 | 23.09.2024 | Blacksea Anchovy | Sunflower oil | Salt | 110 gr | Aluminum |
| D-D-7 | 19-10-2024 | Mackerel | Olive oil | Salt | 110 gr | Aluminium |
| D-D-8 | 27.01.2022 | Norwegian Salmon | Olive oil | Lemon Water, Salt | 85 gr | C/PP |
| D-D-9 | 20.10.2022 | Skipjack | Sunflower oil | Salt | 120 gr | C/PP |
| D-D-10 | 18-03-2023 | Yellowfin Tuna | None | Water | 120 gr | C/PP |
| D-D-12 | 25.09.2023 | Skipjack | Olive oil | Salt | 185 gr | Glass |
| D-DE-1 | 20861E | Tuna | Sunflower oil | Salt | 160 gr | Can |
| D-MI-1 | 20860E CO | Tuna | Sunflower oil Canola oil | Salt | 160 gr | Can |
| D-W-1 | 20861E CO | Tuna | Sunflower oil | Salt | 160 gr | Can |
| I-M-1 | 25 | Tuna | None | Water, salt | 80 gr | Can |
| I-M-2 | 26 | Skipjack | Sunflower oil (%25) | Water, salt | 80 gr | Can |
| I-M-3 | 46 | Yellowfin Tuna | Olive oil (%25) | Water, salt | 160 gr | Can |
| I-Y-1 | 20 | Tuna | Sunflower oil (%25) | Water, Salt | 104 gr | Can |
| I-Y-2 | 14 | Mackerel | Sunflower oil | Water, Salt | 160 gr | Aluminum |
| K-SF-1 | 0265 | Tuna | Olive oil (%25) | Water, salt | 75 gr | Can |
| K-SF-2 | 0051 | Tuna | Sunflower oil (%27) | Water,salt | 80 gr | Can |
| K-SF-3 | 0197 | Tuna | Sunflower oil | Water,salt | 80 gr | Can |
| K-SF-4 | 7339 | Tuna | None | Water,salt | 80 gr | Can |
| K-V-1 | 0346 | Tuna | Sunflower oil | Water, Salt | 160 gr | Can |
| O-F-1 | L1720 | Tuna | Sunflower oil | Salt | 160 gr | Can |
| P-P-1 | 9283 | Tuna | None | Water, salt | 160 gr | Can |
| P-P-2 | 20842E | Tuna | Sunflower oil (%27) | Water, salt | 80 gr | Can |
| S-SAS-1 | 302507280 | Salmon | Sunflower oil | Water,salt | 160 gr | Can |
| S-SAS-2 | 435375812 | Tuna | None | Water, Salt | 160 gr | Can |
| S-SAS-3 | 3021078T3 | Tuna | Olive oil | Salt | 160 gr | Can |
| S-SAS-4 | 406009812 | Tuna | Sunflower oil | Salt,Water | 80 gr | Can |
| T-T-1 | 17848E | Tuna | Sunflower oil | Salt | 80 gr | Can |

prepared solution was added to each beaker containing 50 g of samples. The beakers were covered with aluminum foil and kept at 60 °C for one week until the complete digestion of fish flesh. After all organic material was dissolved entirely, the solution was transferred to the separation funnel, and 500 ml (5 M 1.6 g/mL density) NaI solution was added to the samples. After waiting for one day for density separation, all settled material was removed. The supernatant was transferred to a separate sterile beaker and filtered through a cellulose

membrane filter paper with a 0.45 μm pore size. Then, filter papers were placed in clean petri dishes and removed for microscopic and spectroscopic analyses.

The shapes (fiber/filament and fragment) and colors of all MP-like particles were examined using a camera (EOS 450D, Canon Co., Tokyo, Japan) and attached stereo microscope (SZX16, Olympus Co., Tokyo, Japan). Afterward, the size of MPs was analyzed via ImageJ v1.52s (http://imagej.nih.gov/ij) software. The Feret diameters were used for the measurement of MPs. Size classification was further done as described by *GESAMP (2019)*.

During the microscopic examination, 293 MP-like particles were counted (including replicates), and 79 randomly selected particles were put through μ-Raman analysis (approximately 30% of total particles). MP-like particles were examined using a confocal Raman microscopy system (inVia Qontor, Renishaw, Gloucestershire, UK) with a 532 nm and 785 nm laser. With a connected Leica microscope, the particles were focused at 50× magnification, and 10 s and two accumulation scans were taken with a variable grating setting between 600 l/mm and 1,200 l/mm and a spectrum width of 300–3,200. The spectrums were compared to the ST-Japan Microplastics Library. A 70% or higher match was used as a basis to accept the polymer type's determination.

According to *Gwinnett & Miller (2021)*, common methods for reducing procedural contamination include wearing non-synthetic polymer clothing, cleaning all equipment and surfaces before use, avoiding plastic equipment, and working under controlled air environments. Moreover, taking of controls and blanks during the sampling stage is also a method for measuring contamination. In this study, the common methods of preventing and measuring contamination, which is also mentioned by *Gwinnett & Miller (2021)*, were applied. For this purpose, all equipment was washed three times with microfiltered water and run through an acetone bath before and after use to avoid potential contamination (*Beer et al., 2018*). All equipment was stored in a closed cabinet throughout the study. All solutions used were filtered with GF/C Whatman filter paper with a pore size of 1.2 μm before use. All analyses were performed in a closed laminar flow cabinet (Class-4, Esco Technologies Inc.), and all equipment was covered with aluminum foil during all procedures throughout the study. A triplicate negative control group was prepared to determine if any contamination was present. All procedures for the analysis of the samples were also applied to the control group. In addition, a petri dish was left open for the duration of the analysis, placed under the microscope, and then checked for any possible environmental contamination in the laboratory to identify potential environmental contamination. After examining the control group, two black fiber/filament type particles (0.67 item/petri) were detected in the control petri dishes applied for the digestion, separation, and filtering stages, and no particles were detected in the other petri dishes. While giving the concentration determined for canned fish in this study, the background concentration was subtracted from the overall average.

The human health risks posed by MPs are causes of concern. Recent evidence that shows MPs in human blood, lung, feces, and even human placenta makes estimating human exposure to MPs through food consumption crucial. Although there is no reliable data on canned fish consumption data, the data provided by industry via mainstream media (200 gr per year; *Durdak, 2020*) were used to estimate human exposure to MPs through canned fish consumption per capita for Turkish people (0.54 g/person/week). Additionally, one more

approach was used to assess human exposure to MPs through canned fish consumption. The second approach was made according to data from the Turkish Statistical Association (*TUIK, 2020*) regarding the fish consumption rate per capita for Turkish people (16.84 g/person/week). In this approach, the exposure frequencies represent exposure frequency, with 52, 156, and 260 days per year for exposures of one, three, and five days a week (*Köşker et al., 2022*). The mean intake was calculated using the equation provided by *Akhbarizadeh et al. (2020)* for the canned fish consumption rate:

$$MICF_{52} = C_{CF} * MPs * 52$$
$$MICF_{156} = C_{CF} * MPs * 156$$
$$MICF_{260} = C_{CF} * MPs * 260$$

and fish consumption rate:

$$MIF_{52} = C_F * MPs * 52$$
$$MIF_{156} = C_F * MPs * 156$$
$$MIF_{260} = C_F * MPs * 260$$

where C_CF is the canned fish consumption rate (g/meal) or (g/day/capita), C_F is the fish consumption rate (g/meal) or (g/day/capita), MPs is the MPs number (items/g), MICF is the intake of microplastics based on canned fish consumption, and MIF is the intake of microplastics based on fish consumption (*Akhbarizadeh, Moore & Keshavarzi, 2019*; *Barboza et al., 2020*).

The level of MP was given as MP 100 $g^{-1}$. The Kolmogorov–Smirnov and Shapiro–Wilk tests were applied to determine whether the number of MPs fits a normal distribution, and, if necessary, a proper transformation (logarithmic transformation) was used. One-way ANOVA was done to determine the difference in MP levels between producers, package type, oil type, water usage, and fish species (analysis was done using transformed data) separately. A Pearson correlation analysis was applied to test the relationship between the weight of the samples and MP abundance. Monte Carlo simulations were used to calculate the uncertainty of the distributions of MPs intake assessment through 10,000 iterations. The distribution parameters for the Monte Carlo simulation were determined with the help of MP intake formulas based on fish consumption. The values in the data were used for fitting the distribution parameters (min, max, mean, std error). Accordingly, the simulation process was carried out under normal distribution assumption. All analyses were done using the Statistical Package for the Social Sciences (SPSS) v22 (IBM Co., Armonk, NY, USA) and Tableau v10.2 (Tableau Software LLC., Mountain View, CA, USA) ($p < 0.05$).

## RESULTS

A total of two black fiber/filament type particles (0.67 item/petri) were found in the control petri dishes and procedural blanks. Hence, these numbers are subtracted from the general mean. Out of the 33 investigated canned fish brands produced by seven producers, all had potential MPs in the fish after visual investigation. After visual quantification, all filters
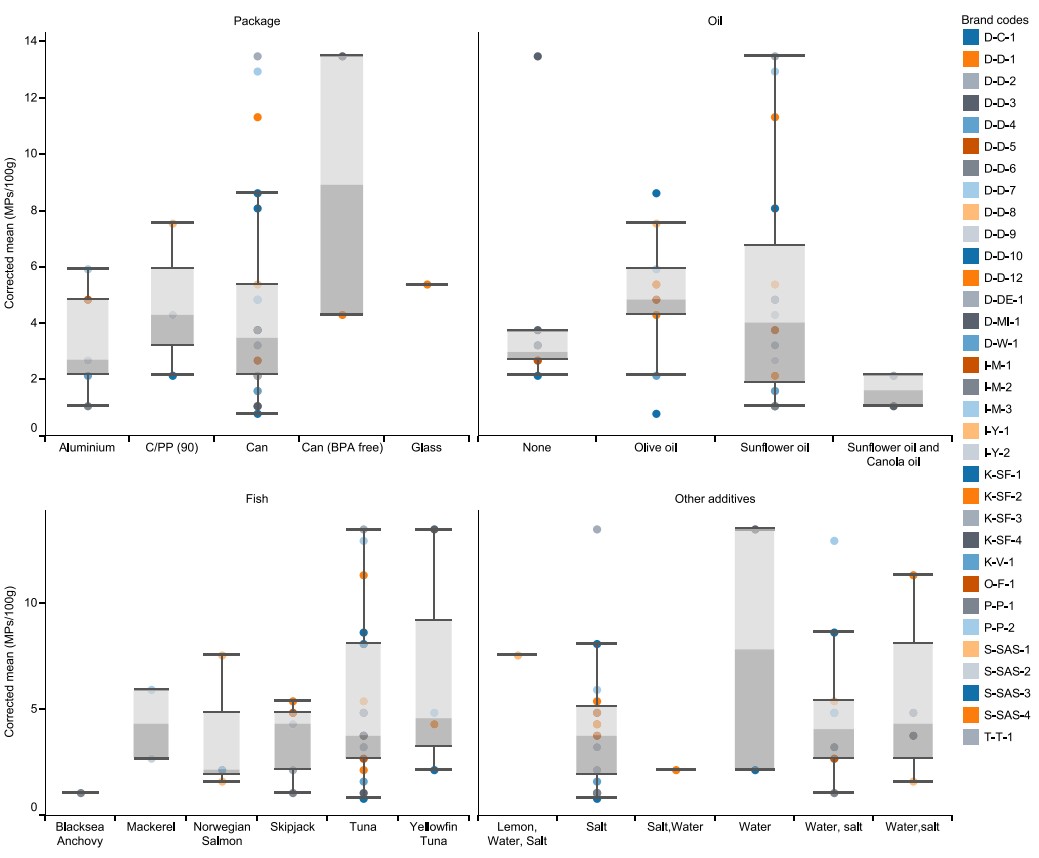

**Figure 1** The number of extracted microplastics per brand and their distribution among package type, fish species, additive oils and other additives.

were subjected to μ-Raman analysis to confirm the detected MPs from the previous works and also distinguish the polymer types. Then, μ-Raman analysis was used, and 79 randomly selected particles were analyzed, and 64 were verified as MPs (81% of the particles were defined as MPs). Hence, this percentage was also used as a correction for the total counted particles (Raw data provided as supplementary material).

After excluding the verified non-MP particles, the mean abundance was estimated at $4.12 \pm 0.62$ MPs /100 g (Table 2; Fig. 1). As can be seen in Table 2, Yellowfin tuna fish in the water (brand D-D-3) and tuna fish in sunflower oil and salt had the highest (mean 13.50 MPs/100 g), and tuna fish in olive oil and salt (brand S-SAS-3) had the lowest (mean 0.81 MPs/100 g) MPs abundance (Table 2; Fig. 1).

There were no statistically significant differences among different package types, different fish species, different oil content, and samples with various additives (water and salt) (One way ANOVA, $p > 0.05$). However, there was a significant difference among different producers (One way ANOVA, $p < 0.05$). Although there was no statistically significant difference between the amounts of MPs according to the packaging types, the highest MPs were found in canned fish labeled as BPA-free (8.91 MPs/100 g; range: 4.3–13.5; std error:

**Table 2  The mean and the corrected mean concentration of microplastics per 100 g of fish flesh and the MICF (intake of canned fish) and MIF (intake of fish) rates for canned fish consumers.**  The exposure frequencies are 52, 156, and 260 days of consumption per year.

| Brand | MPs/100 g | Corrected mean ± Std Error (MPs/100 g) | MICF | | | MIF | | |
|---|---|---|---|---|---|---|---|---|
| | | | 52 days per year | 156 days per year | 260 days per year | 52 days per year | 156 days per year | 260 days per year |
| D-C-1 | 10.00 | 8.10 ± 1.53 | 0.32 | 0.97 | 1.62 | 10.13 | 30.40 | 50.66 |
| D-D-1 | 5.33 | 4.32 ± 0.33 | 0.17 | 0.52 | 0.87 | 5.40 | 16.21 | 27.02 |
| D-D-2 | 2.67 | 2.16 ± 0.33 | 0.09 | 0.26 | 0.43 | 2.70 | 8.11 | 13.51 |
| D-D-3 | 16.67 | 13.50 ± 2.33 | 0.54 | 1.62 | 2.71 | 16.89 | 50.66 | 84.44 |
| D-D-4 | 2.67 | 2.16 ± 1.33 | 0.09 | 0.26 | 0.43 | 2.70 | 8.11 | 13.51 |
| D-D-5 | 6.00 | 4.86 ± 0.58 | 0.19 | 0.58 | 0.97 | 6.08 | 18.24 | 30.40 |
| D-D-6 | 1.33 | 1.08 ± 0.33 | 0.04 | 0.13 | 0.22 | 1.35 | 4.05 | 6.76 |
| D-D-7 | 7.33 | 5.94 ± 1.20 | 0.24 | 0.71 | 1.19 | 7.43 | 22.29 | 37.15 |
| D-D-8 | 9.33 | 7.56 ± 1.76 | 0.30 | 0.91 | 1.52 | 9.46 | 28.37 | 47.29 |
| D-D-9 | 5.33 | 4.32 ± 0.88 | 0.17 | 0.52 | 0.87 | 5.40 | 16.21 | 27.02 |
| D-D-10 | 2.67 | 2.16 ± 0.88 | 0.09 | 0.26 | 0.43 | 2.70 | 8.11 | 13.51 |
| D-D-12 | 6.67 | 5.40 ± 0.88 | 0.22 | 0.65 | 1.08 | 6.76 | 20.27 | 33.78 |
| D-DE-1 | 4.00 | 3.24 ± 0.58 | 0.13 | 0.39 | 0.65 | 4.05 | 12.16 | 20.27 |
| O-F-1 | 4.67 | 3.78 ± 1.86 | 0.15 | 0.45 | 0.76 | 4.73 | 14.19 | 23.64 |
| I-M-1 | 3.33 | 2.70 ± 1.20 | 0.11 | 0.32 | 0.54 | 3.38 | 10.13 | 16.89 |
| I-M-2 | 1.33 | 1.08 ± 0.33 | 0.04 | 0.13 | 0.22 | 1.35 | 4.05 | 6.76 |
| I-M-3 | 6.00 | 4.86 ± 1.15 | 0.19 | 0.58 | 0.97 | 6.08 | 18.24 | 30.40 |
| D-MI-1 | 1.33 | 1.08 ± 0.33 | 0.04 | 0.13 | 0.22 | 1.35 | 4.05 | 6.76 |
| P-P-1 | 4.00 | 3.24 ± 1.15 | 0.13 | 0.39 | 0.65 | 4.05 | 12.16 | 20.27 |
| P-P-2 | 16.00 | 12.96 ± 3.21 | 0.52 | 1.56 | 2.60 | 16.21 | 48.64 | 81.06 |
| S-SAS-1 | 2.00 | 1.62 ± 0.58 | 0.06 | 0.19 | 0.32 | 2.03 | 6.08 | 10.13 |
| S-SAS-2 | 3.33 | 2.70 ± 0.67 | 0.11 | 0.32 | 0.54 | 3.38 | 10.13 | 16.89 |
| S-SAS-3 | 1.00 | 0.81 ± 0.50 | 0.03 | 0.10 | 0.16 | 1.01 | 3.04 | 5.07 |
| S-SAS-4 | 2.67 | 2.16 ± 1.33 | 0.09 | 0.26 | 0.43 | 2.70 | 8.11 | 13.51 |
| K-SF-1 | 10.67 | 8.64 ± 1.76 | 0.35 | 1.04 | 1.73 | 10.81 | 32.43 | 54.04 |
| K-SF-2 | 14.00 | 11.34 ± 3.61 | 0.45 | 1.36 | 2.27 | 14.19 | 42.56 | 70.93 |
| K-SF-3 | 6.00 | 4.86 ± 1.15 | 0.19 | 0.58 | 0.97 | 6.08 | 18.24 | 30.40 |
| K-SF-4 | 4.67 | 3.78 ± 0.67 | 0.15 | 0.45 | 0.76 | 4.73 | 14.19 | 23.64 |
| T-T-1 | 16.67 | 13.50 ± 0.88 | 0.54 | 1.62 | 2.71 | 16.89 | 50.66 | 84.44 |
| K-V-1 | 6.00 | 4.86 ± 1.53 | 0.19 | 0.58 | 0.97 | 6.08 | 18.24 | 30.40 |
| D-W-1 | 2.00 | 1.62 ± 0.58 | 0.06 | 0.19 | 0.32 | 2.03 | 6.08 | 10.13 |
| I-Y-1 | 6.67 | 5.40 ± 1.86 | 0.22 | 0.65 | 1.08 | 6.76 | 20.27 | 33.78 |
| I-Y-2 | 3.33 | 2.70 ± 0.33 | 0.11 | 0.32 | 0.54 | 3.38 | 10.13 | 16.89 |
| Grand Mean | 5.93 | 4.12 ± 0.62 | 0.17 ± 0.02 | 0.50 ± 0.07 | 0.83 ± 0.12 | 5.15 ± 0.78 | 15.46 ± 2.34 | 25.77 ± 3.9 |

**Notes.**
MICF,  Intake of microplastics based on canned fish consumption;  MIF,  intake of microplastics based on fish consumption.

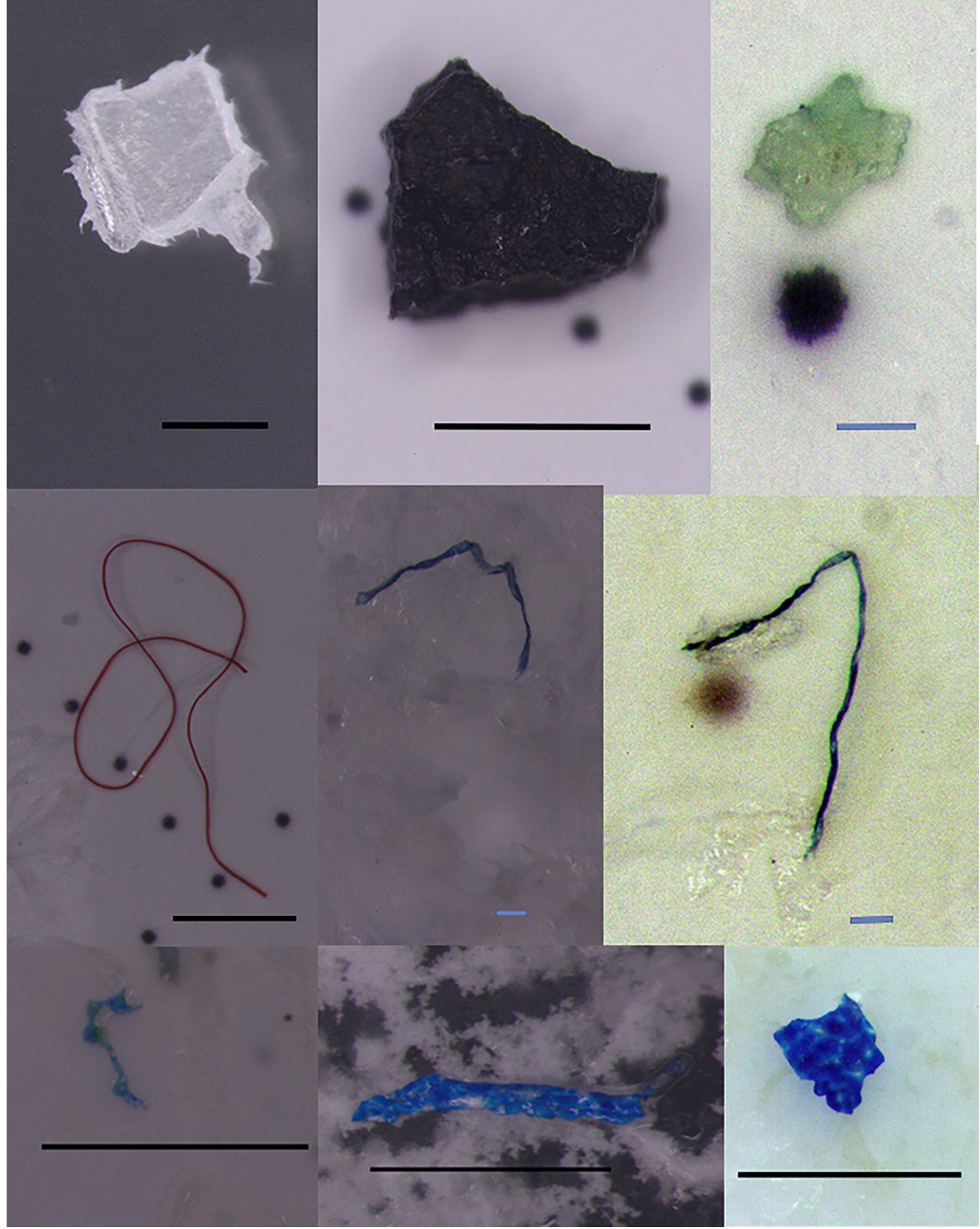

**Figure 2** Extracted microplastics with different colors and shapes. The black bars represent one mm scale, and the blue bars represent 0.1 mm.

4.60), and the lowest was found in aluminum canned fish (3.35 MPs/100 g; range:0.8–13.5; std error: 0.81) (Fig. 2).

Regarding the shape of the particles, the share of fragments was 57.3%, and the share of fibers was 42.7% (Fig. 3). The particle size ranged between 0.06 and 5.14 mm for the fragments and between 0.27 and 5.89 mm for the fibers (Fig. 3). The color of MPs in canned
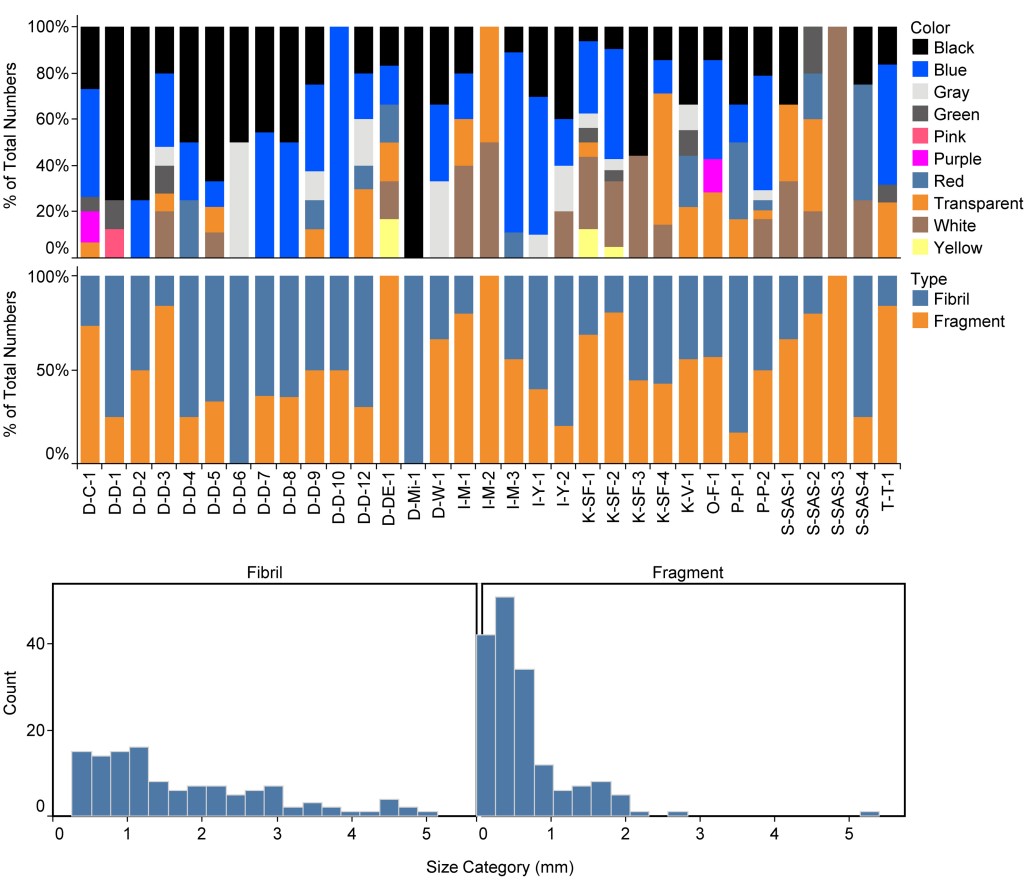

**Figure 3** Color (upper panel), shapes (middle panel), and the size distribution (lower panel) of identified microplastics in the canned fish samples.

fish was as follows: blue (34.8%) >black (27.3%) >white (11.9%) >transparent (10.6%) (Fig. 3).

Twelve synthetic polymers and two natural polymers (cellulose and chitin) were identified via μ-Raman (Table 3). The most abundant polymers were polyolefine (21.8%), polyacrylonitrile (10.9%), and followed by Poly(methacrylic acid methyl ester) (9.38%), polyamide (7.8%), polyethylene terephthalate (7.8%), and polypropylene (7.8%).

MPs in canned seafood can be considered an emerging risk to human health. Hence, their transfer into the human diet should be monitored carefully. This study considered two different fish consumption ratios to understand the risk posed by MPs in canned fish to humans. The first one is the rate of canned fish consumption per capita for Turkish people and the second one is the general seafood consumption rate of Turkish people. It should be noted that the canned fish consumption rate was taken from a speech given by industry representatives, and the general seafood consumption rate was taken from *TUIK (2020)*. Hence, the intake of MP items through canned fish, based on canned fish consumption (MICF), was estimated at $0.17 \pm 0.02$ items/g/year (once a week), $0.50 \pm 0.07$ items/g/year (three times a week), and $0.83 \pm 0.12$ items/g/year (five times a week), for

**Table 3** The chemical composition of extracted particles.

| Type | Polymer | % |
|---|---|---|
| Fiber | Chitin | 18.18 |
| | Poly(vinyl acetate) | 18.18 |
| | Polyamide - Nylon 6, 6 | 18.18 |
| | Cellulose | 9.09 |
| | Ethylene-vinyl acetate | 9.09 |
| | Polyacrylonitrile | 9.09 |
| | Polypropylene | 9.09 |
| | Polyvinylidene chloride | 9.09 |
| Fragment | Polyolefin | 26.42 |
| | Poly(methacrylic acid methyl ester) | 11.32 |
| | Polyacrylonitrile | 11.32 |
| | Polyethylene Terephthalate | 9.43 |
| | Polyethylene | 7.55 |
| | Polypropylene | 7.55 |
| | Polystyrene | 7.55 |
| | Polyamide - Nylon 6, 6 | 5.66 |
| | Polyvinylidene chloride | 5.66 |
| | Epoxy | 3.77 |
| | Poly(vinyl acetate) | 3.77 |
| Total | Polyolefin | 21.88 |
| | Polyacrylonitrile | 10.94 |
| | Poly(methacrylic acid methyl ester) | 9.38 |
| | Polypropylene | 7.81 |
| | Polyethylene Terephthalate | 7.81 |
| | Polyamide - Nylon 6, 6 | 7.81 |
| | Polyvinylidene chloride | 6.25 |
| | Polystyrene | 6.25 |
| | Polyethylene | 6.25 |
| | Poly(vinyl acetate) | 6.25 |
| | Epoxy | 3.13 |
| | Chitin | 3.13 |
| | Ethylene-vinyl acetate | 1.56 |
| | Cellulose | 1.56 |

Turkish consumers. The intake of MP items through canned fish, based on seafood consumption (MIF), was estimated at $5.15 \pm 0.78$ items/g/year (once a week), $15.46 \pm 2.34$ items/g/year (three times a week), and $25.77 \pm 3.9$ items/g/year (five times a week), for Turkish consumers (Table 2; Fig. 4). The Monte Carlo simulation was used to estimate the risk for consumers related to the consumption of canned fish. Monte Carlo simulation was applied to estimate the uncertainty of the results. Hence this allows us to know the variability and the quality of measurements. Moreover, it provides a probability distribution in terms of statistical indicators (percentiles) and, thus, evidence of the probability of overcoming a specified reference threshold. The Monte Carlo simulation was performed on all MP data.
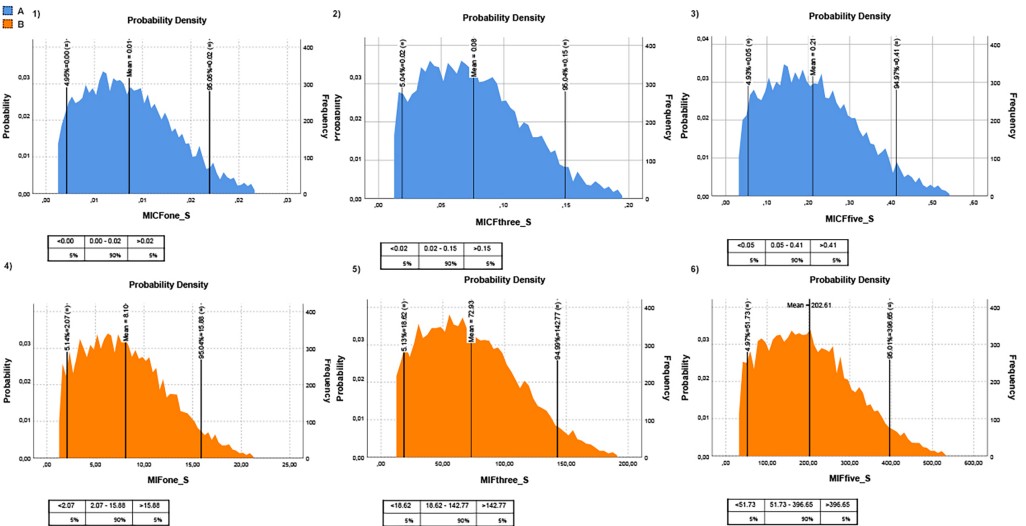

**Figure 4   Monte-Carlo simulation results for MPs intake for Turkish people.** (A) MPs intake risk based on canned fish consumption (blue graph); (B) MPs intake risk based on fish consumption (orange graph). The lower panels from 1 to 3 represent one, three, and five times the consumption of canned fish per week, respectively; the lower panels 4 to 6 represent one, three, and five times consumption of fish per week, respectively. The tables below each graph show MPs intake levels according to percentage. The in-take unit is MP/g/year.

The probabilistic distributions of MPs intake levels (MICF and MIF) obtained from Monte Carlo simulations are shown in Fig. 4, and the descriptive statistical values, including the 5th percentile, mean and 95th percentile. The exposure time was considered one day/three days/five days per week. All the values of MICF were lower than 1 MPs/person/year, and MIF values were higher than 1 MPs/person/year (Fig. 4). Despite no legislation establishing the limits of total MP in seafood, the health risks related to plastics and associated chemicals raise concern.

## DISCUSSION

Canned fish analyzed in this study showed a similar occurrence of MPs concentration among and within the brands and producers; this similarity suggests that MPs contamination of canned fish is a common contamination problem. Unfortunately, there is not much similar research to compare the obtained results for the Turkish market. However, some previous studies in other countries confirm the occurrence of MPs in canned fish. For instance, *Karami et al. (2018)* investigated 20 brands purchased from Australian and Malaysian markets and identified MPs in four of 20 samples. *Akhbarizadeh et al. (2020)* investigated fifty canned fish (tuna and mackerel) samples from seven brands purchased from Iranian hypermarkets. They reported MPs in 80% of the samples. In a more recent study, *Diaz-Basantes et al. (2022)* investigated 32 different cans belonging to four brands of canned tuna purchased from Ecuador. They reported MP occurrence in all of the analyzed samples. The possible reasons for the difference between the results of

our study and the mentioned studies might be due to the production conditions, species diversity, additives (oil, salt, water, *etc.*), and analysis methods.

Although previous studies reported PET, PP, and PS as the most commonly identified polymers in aquatic organisms and canned seafood (*Karami et al., 2018*; *Koongolla et al., 2020*; *Akhbarizadeh et al., 2020*; *Diaz-Basantes et al., 2022*), this pattern was identified in this study. The most identified polymer type in this study was found as polyolefin. The polymer pattern determined in this study indicates the packaging used in canning is a potential source of contamination.

For preservation, canning fish requires using a container impervious to liquids, gases, and microorganisms and using adequate heat treatment to neutralize microorganisms that can grow under cold chain-free storage conditions (*Ababouch, 2002*). Although the canned materials used vary, mostly metal materials with an inner surface coated with a coating polymer are used (*Geueke, 2017*). In addition, composite packages made of different materials of different layers and glass packages are also used. Coatings of cans can also be polymers. Many different can coatings are commercially available. Coatings materials contain various agents, lubricants, adhesives, pigments, *etc.* (*Geueke, 2017*). Epoxy-based coatings have the highest market share at more than 90%. However, can manufacturers have started to replace epoxy coatings with alternatives such as acrylic, polyolefin, and polyester as a consequence of increasing public discussions and recent regulatory decisions (*Oldring & Nehring, 2007*; *Driffield et al., 2018*). This situation poses the risk of these plastic coatings breaking down and contaminating the food, both during the production of the packaging and during the canning of the fish or opening the can while consuming. Our results show that most of the polymer types in this study are compatible with the polymer types used in can coating. Therefore, based on the results of this study, it is possible to say that the can itself primarily causes MPs in canned fish.

Other polymer types detected with μ-Raman reveal that packaging is not the only source of MPs contamination in canned fish. For example, fibre-type plastics can originate both during production and from the ecosystem where fish are caught. It is widely known that fibre-type plastics mostly originate from textile products and are commonly found in the marine ecosystem (*Balasaraswathi & Rathinamoorthy, 2022*) and the indoor air (*Chen et al., 2022*; *Kashfi et al., 2022*). Therefore, both the fish species used for canning and indoor air can be the source of fibre-type microplastics in canned fish. As a matter of fact, the finding of a large number of fibres in various colours and sizes in our results can be considered as an indicator of this.

The occurrence of a high number of fibers can also be a sign of contamination. However, in this study, full contamination prevention protocols were applied. All solutions, including cleaning liquids, were filtered before use, and all the analysis was conducted under a closed flow laminar cabinet. Microscopic examination for the counting particles was conducted under a fully covered microscope. Moreover, all procedures for analyzing the samples were also applied to the control group. In addition, a petri dish was left open for the duration of the microscopic analysis. Finally, identified contamination was subtracted from the mean concentration.

On the other hand, it is thought that the protective clothing worn by the employees may also be a potential source of contamination of MPs, specifically fiber types. Legal regulations for contamination in the production process mostly focus on preventing contamination from microorganisms. There is no regulation on the risk of MPs contamination from protective clothing. Considering that plastic materials (polystyrene, polyethylene, polyvinyl chloride, *etc.*) are mostly used in the production of protective clothing, it will be understood that it is in parallel with the MPs contamination revealed in this study. It is possible to say that especially acrylonitrile type MPs (mostly blue; Fig. 2) may be caused by particles that break off as a result of damage to the gloves worn by the workers during production. The fact that there are many cases on an online platform that compiles the complaints of Turkish consumers in this regard supports this phenomenon (*Sikayetvar, 2022*).

The presence of MPs in the stomach and digestive tracts of edible fish, whether farmed or wild, has been reported in previous studies (*Garcia et al., 2021*). In addition, there is a risk of MPs in farmed fish, both from the environment (*Gomiero et al., 2020*) and from the fish meal (*Gündoğdu et al., 2021*). The canned fish investigated in this study were both wild-caught (*e.g.*, anchovy, mackerel) and farmed (*e.g.*, salmon) fish. Therefore, it can be said that both types of fish production may be a source of MP contamination of canned fish investigated in this study. There are also other possible sources of MPs in canned fish. The additives such as water, oil, and salt can also be potential sources of MPs. *Danopoulos, Twiddy & Rotchell (2020)* stated that drinking water could be considered one possible medium for introducing MPs into the human body. Therefore tap water used during canned fish manufacture can be one of the sources of MPs in canned fish investigated in this study. Similarly, the additives such as salt and oil also contain synthetic polymers, as reported by different researchers (*Gündoğdu, 2018*; *Zhou et al., 2020*).

The presence of MPs in canned fish products can be considered a possible threat to human health (*Gündoğdu et al., 2022a*; *Gündoğdu et al., 2022b*). Hence, their transfer into the human diet should be monitored carefully. This study considered two different fish consumption ratios to understand the risk posed by MPs in canned fish to humans. The first one is the rate of canned fish consumption per capita for Turkish people and the second one is the general seafood consumption rate of Turkish people. It should be noted that the canned fish consumption rate was taken from a speech given by industry representatives, and the general seafood consumption rate was taken from *TUIK (2020)*.

According to the calculations, the intake of MPs for Turkish consumers ranged from 0.03 to 2.71 MPs/year based on the canned fish consumption rate and ranged from 1.01 to 81.44 MPs/year based on the seafood consumption rate. This result is much lower than the findings of *Akhbarizadeh et al. (2020)*. On the other hand, the findings of *Karami et al. (2018)* were similar to the findings of this study. There are many possible reasons for the differences between the human intake values in this study and those in other studies. Human exposure to Ps through food is also directly related to the presence of MPs in canned fish. For example, *Akhbarizadeh et al. (2020)* reported 128 MPs/100 g, and *Diaz-Basantes et al. (2022)* reported 442–692 MPs/100 g. Both estimates are considerably higher than the 1.08–13.5 MPs/100 g reported in this study. The most important reason is the difference in the consumption rate of canned fish or seafood per capita among countries. Although

Türkiye has a coast to the sea, it lags far behind in seafood consumption. This situation reduces the risk of microplastic intake from canned fish.

## CONCLUSIONS

The occurrence of MPs in several other seafoods like seaweed nori (*Li et al., 2020*), stuffed mussels (*Gündoğdu et al., 2021*), and oysters (*Do et al., 2022*) might be a warning of the potential health risks associated with the long-term exposure to seafood derived MPs sources. In addition, the identification of processed seafood products as a potential route of MPs exposure, including the results found in this study, suggests that the extent of the problem is even greater. Although it is thought that the source of MP that cause human exposure through seafood is aquatic environments, this study also reveals that packaging is also an important source of MPs pollution. As a result, it is revealed that the increasing plastic production and the increasing plastic consumption increase severity of the microplastic pollution and continue to create risks in terms of human and environmental health from different sources. These findings necessitate the enactment of legislation that sets the total MP limits in the processing of seafood.

Moreover, a plastics treaty, recently announced by the UN in Kenya, needs to be established by 2024, which should include a global cap on plastic production to end pollution and harm to human health (*Bergmann et al., 2022*).

### Funding

The authors declare that no funds, grants, or other support were received in support of this work. The funders had no role in study design, data collection and analysis, decision to publish, or preparation of the manuscript.

### Competing Interests

The authors declare there are no competing interests.

### Author Contributions

- Sedat Gündoğdu conceived and designed the experiments, performed the experiments, analyzed the data, prepared figures and/or tables, authored or reviewed drafts of the article, and approved the final draft.
- Ali Riza Köşker performed the experiments, analyzed the data, authored or reviewed drafts of the article, collection of samples, and approved the final draft.

### Data Availability

The data is available in the Supplementary File.

### Supplemental Information

Supplemental information for this article can be found online at http://dx.doi.org/10.7717/peerj.14627#supplemental-information.

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
