# Peer review of "Microplastic contamination in canned fish sold in Türkiye"

_PeerJ, doi:10.7717/peerj.14627_

## Round 0.1 · original submission · Major Revisions

The reviewers have determined (and I agree), that your manuscript requires some revisions. The reviewer comments are straightforward, so I will not repeat them here. However, an important point raised by the reviewers is worth emphasizing. There is some uncertainty which should be addressed in your revision with respect to the methods used to quantify such small particle counts; even a small amount of contamination would have a large impact on the outcome. Please address these and the other comments in your revision.

Reviewer 1 ·

Basic reporting

This is a good work. The work presented is very interesting and innovative. The methods used are adequate. The information is presented correctly and understandably.
I consider the manuscript is well written and can be published.
The novelty is scarce but very complete information on the canned fish is found. It has interesting results that may be important

Experimental design

The technical content is correct.

Validity of the findings

Data interpretation is quite interesting and with new conclusions.

Additional comments

The literature cited is complete, as far as I know,
English usage is correct.
Quality of figures is excellent.

Reviewer 2 ·

Basic reporting

257 - This type of text belongs in the discussion, not the results.

The potential issue of contamination using such methods to quantify such small particle counts needs to be sufficiently addressed in text. As well as the wider implications for other studies. This will provide better context for any conclusions made.

Experimental design

Line 141 - It's not clear whether you mean there were three replicates of the 13 brands, or three replicates of the 33 canned fish samples. Please clarify.

170 - Why was 30% chosen as the threshold for polymer validation?

216 - Need to report/define the transformations used? Also, which parts of the collected data were analysed differently, as implied here.

There needs to be a discussion of the potential issues regarding contamination in this study (and others like it) due to the small number of particles being detected, it presents a major issue with such studies that needs to be addressed.

Validity of the findings

244 - The effect size of this difference would help clarify this finding.

246 - Need to provide margins for error here, as if there was no statistical difference this point is challenged.

Additional comments

I thank the authors for their submission. Overall, my suggestions can be summarised with addressing potential error caused by contamination risk, clarify details in the methods (largely in the data analysis), and use effect size to better clarify any significant difference.

Reviewer 3 ·

Basic reporting

The authors write well, although with some errors and lack of clarity in certain places.

Lines 64-65: The sentence, "Microplastics (MPs) are small plastic particles with less than 5 mm dimensions" is redundant with the last sentence. Can be removed.

Lines 73-74: The last part of this sentence needs rewording.

Lines 117-118: There is a loose "e" here.

Lines 118-119: This sentence is unclear.

Line 197: This link, and the other cited in the text, should be properly referenced in the references section. According to PeerJ's reference style, it could then be cited in the text by author and year.

Experimental design

The research is useful for providing data on microplastics contamination in fish products. The methods are mostly clearly explained, with some exceptions:

Lines 215-217: These tests should be applied to the data separated by group in the case of a 1-way ANOVA. What transformations were applied?

Lines 220-221: Please provide more detail on the Monte Carlo simulations. There should be enough information present so that someone else could take your data and carry out the same technique with the same results. For example, it's not clear what "fitting distributions from MP concentrations in canned fish" means mathematically.

Validity of the findings

Lines 264-266: Given the equations in the methods sections, these estimates should represent the number of items consumed over a certain period of days, not items/g/year. If there are so many MPs per g of fish, why would the per g estimates change depending on the intake period? Also, the numbers here seem low. I think something has gone wrong with the calculations.

Otherwise, the findings seem valid.

Additional comments

Table 2: Please define acronyms in the table description.

Figure 4: What are the units for MP intake here? Is it per year?

---

## Round 0.2 · accepted · Accept

Thank you for your efforts to revise your manuscript according to reviewer comments. The reviewers and I believe that your manuscript is now ready for publication.

Reviewer 1 ·

Basic reporting

The manuscript is correct and may be published.

Experimental design

The manuscript is correct and may be published

Validity of the findings

The manuscript is correct and may be published

Additional comments

The manuscript is correct and may be published